# A Normalized Model of a Microelectromechanical Relay Calibrated by Laser-Doppler Vibrometry

**DOI:** 10.3390/mi13101728

**Published:** 2022-10-13

**Authors:** Jessica Marvin, Michael Jang, Daniel Contreras, Matthew Spencer

**Affiliations:** Harvey Mudd College, Claremont, CA 91711, USA

**Keywords:** relay, parallel-plate, normalization, simulation, Verilog-A, laser Doppler vibrometry

## Abstract

This work presents a behavioral model for a microelectromechanical (MEM) relay for use in circuit simulation. Models require calibration, and other published relay models require over a dozen parameters for calibration, many of which are difficult to extract or are only available after finite element analysis. This model improves on prior work by taking advantage of model normalization, which often results in models that require fewer parameters than un-normalized models. This model only needs three parameters extracted from experiment and one dimension known from device fabrication to represent its non-contact behavior, and two additional extracted parameters to represent its behavior when in contact. The extracted parameters–quality factor, resonant frequency, and the pull-in voltage–can be found using laser Doppler vibrometry. The device dimension is the actuation gap size, which comes from process data. To demonstrate this extraction process, a series of velocity step responses were excited in MEM relays, the measured velocity responses were used to calibrate the model, and then then simulations of the model (implemented in Verilog-A) were compared against the measured data. The error in the simulated oscillation frequency and peak velocity, two values selected as figures of merit, is less than 10% across many operating voltages.

## 1. Introduction

Vertical-gap microelectromechanical (MEM) relays have wide ranging applications in circuit design, especially in radio frequency applications [1]. For instance, MEM relays have been used as transmit/receive switches and antenna tuners in radios [2]. Recent work has also explored the application of MEM relays to digital circuit design, examining their potential as logic gates [3], power gates [4], and back-end-of-line integrated switches [5,6].

Circuit designers rely on behavioral models of MEM relays, often written in SPICE or Verilog-A, to design relays using electronic design automation software [3,7,8,9,10], but these models often require the user to specify many material and device parameters, which can limit their use. For instance, the model in [3] requires two material properties, nine layout dimensions, two vertical device fabrication dimensions, and three values that are extracted from measurements. As another example, the model in [8] requires six material parameters, two layout dimensions three vertical device fabrication dimensions, and one parameter extracted from finite element analysis (FEA). Models that have many parameters or difficult-to-extract parameters, particularly if the parameters require FEA or device measurements, are more difficult to develop.

Model normalization simplifies relay models, resulting in fewer, easier-to-extract parameters. Normalizing models is a technique that was originally developed for circuit simulation to reduce rounding errors [11]. The technique requires dividing a behavioral model by an upper bound to change the scale of the numbers representing the model’s state [12]. Properly selected normalization terms can also reduce the number of parameters in a model [13]. No normalized relay models have been reported to date.

Normalized models still have parameters, and this work suggests that laser Doppler vibrometry (LDV) is a good tool to measure the model parameters the normalized model in this work. This is because typical LDV characterization captures the dynamic behavior of a device from its step or harmonic response, making it well-suited to directly measure quality factor and natural frequency. This is not a new observation because LDV is a proven technique for extracting models from microdevices [14,15], and LDV has also been used for switch models in RF switches [16,17]. However, the RF switch models in [16,17] rely on FEA models in conjunction with LDV. Combining LDV and normalized analytical models offers the prospect of directly measuring relay model parameters and eliminating the need for FEA.

This work contributes a normalized dynamic model of a vertical-gap MEM relay that uses a small number of easily extracted parameters in the behavioral model. The normalized model relies on three parameters that can be extracted from measurements—quality factor, $Q_{0}$, natural frequency, $\omega_{0}$, and pull-in voltage, $V_{pi}$—and one vertical device fabrication dimension—the actuation gap, *g*—to describe the relay’s motion when the drain does not contact the source. The model can be extended to describe contact by adding two more extracted parameters and a few tuning parameters to affect simulator behavior. All parameters are calibrated using only simple LDV measurements, and calibrating them does not require FEA. This model was validated by implementing it in Verilog-A, simulating it using a standard circuit simulator (hSpice) and comparing the simulated velocity against measurements.

Further description of this model can be found in the remainder of this work. Section 2, describes the device being modeled, the normalized model, the experiments used to extract model parameters, and the algorithm for extracting model parameters from measured data. Section 3 shows the measured data, the extracted parameters, and the comparison of the simulated model to measurements. Section 4 concludes the paper and suggests future work.

## 2. Materials and Methods

### 2.1. Device Description

The relay analyzed in this work was first demonstrated in [18], and a micrograph and cross-section of a device are shown in Figure 1. The device is a parallel-plate MEM switch in which a movable shuttle, called the gate, is supported by folded flexures above an electrode on the substrate, called the body. The gate and the body are separated by a distance *g* = 220 nm, called the gap.

During operation, a Voltage $V_{g}$ is applied to the gate and a different Voltage $V_{b}$ is applied to the body. The difference between these Voltages, $V_{gb}=V_{g}-V_{b}$, creates an electrostatic force that causes the gate to move toward the body. After the device has moved a distance $g_{d}$ = 22 nm, called the dimple gap, an electrode affixed to the gate, called the drain, makes contact with an electrode affixed to the substrate, called the source. A drain-source current, $I_{ds}$, can flow when the drain and source are in contact. No current flows between the gate and the drain because they are separated by a non-conductive oxide. The Voltage at which the source adn the drain make contact is called the contact Voltage, $V_{con}$.

The displacement of this parallel plate actuator from its resting position, *x*, is determined by a balance of electrostatic forces from $V_{gb}$ and spring forces from the supporting flexures. This electrostatic force balance results in the well-known pull-in instability: if x>g/3, the electrostatic force increases with decreasing gap size faster than the flexures’ restoring force. This instability occurs at the pull-in voltage, $V_{pi}$. The devices in this paper have $g_{d}<g/3$, so the drain electrode prevents the gate from displacing to g/3. As a result, the relay always operates in non-pull-in mode. However, it is still possible to calculate $V_{pi}$ for a non-pull-in device–$V_{pi}$ is the Voltage that would be required to displace the gate by g/3 if drain electrode were not present—and doing so is important in the following analysis.

Note that using $g_{d}$ = 22 nm is a deviation from [18], which lists the as-designed $g_{d}$ as 60 nm. All of the measured $V_{con}$ values in this work, which are approximately 14 V, are consistent with $g_{d}$ = 22 nm. Changing $g_{d}$ affects the value of $V_{con}$ without changing $V_{pi}$ or other parameters, all of which appear correct because they were derived both experimentally and theoretically. Changing $g_{d}$ made the data consistent with the least possible change to published process parameters.

### 2.2. Free Space Model

The relay can be modeled as a second order dynamical system when it is not in contact with the drain and source electrodes,
(1)$$m\ddot{x}+b\dot{x}+kx=\frac{1}{2}\epsilon_{0}V_{gb}^{2}\frac{A}{{\left(g-x\right)}^{2}},$$
where *m* is the effective mass of the relay, *b* is the damping coefficient of the relay, *k* is the spring constant of the folded flexures, and *x* is the displacement of the gate. The right hand side of the equation is an electrostatic force, denoted $F_{el}$ in future equations, where $\epsilon_{0}$ is the relative vacuum permittivity, assumed to be 8.85 pF/m, $V_{gb}$ is the voltage between the gate and the body, *A* is the active electrode overlap area between the gate and body, and *g* is the size of the gap between the gate and body.

The expression for the dynamic system can be normalized by kg, to give
(2)$$\frac{\ddot{\tilde{x}}}{\omega_{0}^{2}}+\frac{\dot{\tilde{x}}}{\omega_{0}Q_{0}}+\tilde{x}=\frac{4}{27}\frac{V_{gb}^{2}}{V_{pi}^{2}}\frac{1}{{\left(1-\tilde{x}\right)}^{2}},$$
where $\tilde{x}=x/g$ is the normalized gap size, $\omega_{0}=\sqrt{k/m}$ is the system’s natural frequency, $Q_{0}=\sqrt{mk}/b$ is the system’s quality factor, and $V_{pi}=\sqrt{\frac{8}{27}\frac{kg^{3}}{\epsilon_{0}A}}$ is the pull-in Voltage of a parallel plate actuator. This model is modified in Section 2.5 to describe relay behavior in contact with the surface, and additional practical details required to implement this model are discussed in Appendix A.

### 2.3. Experiment

Calibrating and verifying the model in Section 2.2 required measurements of the MEM relay’s dynamic behavior. Those measurements were obtained by configuring the relay as a pseudo-inverter, a common test structure [3], and manipulating the voltages on the device to induce a series of mechanical step responses. The velocity of the gate during these step responses was measured with LDV.

The details of the experimental configuration are as follows: the drain was connected to a 2 V Voltage source through a 47 kΩ pull-up resistor, the source was grounded, the body was connected to a bias Voltage that varied from 6.5 to 13.0 V in different trials, and the gate was driven by a ±1 V square wave. This resulted in small displacements of the device, and the velocity corresponding to those displacements was measured using LDV—specifically, a Polytec OFV-534 sensor head with an OFV-5000 controller and a VD-09 velocity decoder. The velocity decoder was set to a resolution of 200 mm/s/V, so it had a bandwidth of 2.5 MHz. A 10x lens on the OFV 534 sensor head was used to focus the laser spot onto the gate electrode to measure micromechanical movement. The laser spot was adjusted to achieve maximum focus on the device gate. A velocity response of the gate to a step input in $V_{g}$ and the displacement curve calculated by integrating the velocity are shown in Figure 2.

The Voltage ranges used in this experiment were selected deliberately. The swing value of $V_{g}$ was set to minimize stress on the gate oxide and to create small displacements of the gate, which allowed for trials to be carried out at many different $V_{b}$ biases. The minimum $V_{b}$ bias was chosen to make the gate move quickly enough to overcome the LDV noise floor. The maximum $V_{b}$ was limited to prevent the drain from contacting the source. Though the drain and source make contact in normal operation, avoiding contact here was desirable for two reasons: contact introduces a risk of device failure into an otherwise non-destructive test, and contact introduces non-linear forces that made it harder to extract model parameters from the measured velocity.

### 2.4. Extracting Model Parameters

Equation (Equation 2) is described by four parameters: the pull-in voltage, the natural frequency, the quality factor, and the gap size. Though the gap size does not appear directly in (Equation 2), it is necessary to normalize the *x* variable. The first three parameters can be extracted experimentally, while the gap size requires knowledge of the fabrication process.

In order to find $V_{pi}$, two states of static equilibrium can be compared before and after the Voltage step on the gate. Several values are known in these static equilibria: before the gate moves, the system has a gate-body voltage called $V_{gb,open}$, after the gate moves to its next position the gate-body voltage is called $V_{gb,closed}$, and velocity and acceleration are both zero in both static equilibria. These known values were substituted into Equation (Equation 2), which resulted in the following system of equations in $V_{pi}$, $\tilde{x}$, and a small shift in the normalized displacement, $\Delta \tilde{x}$:(3)$$\tilde{x}=\frac{4}{27}{\left(\frac{V_{gb,open}}{V_{pi}}\right)}^{2}\frac{1}{{\left(1-\tilde{x}\right)}^{2}},$$
(4)$$\tilde{x}+\Delta \tilde{x}=\frac{4}{27}{\left(\frac{V_{gb,closed}}{V_{pi}}\right)}^{2}\frac{1}{{\left(1-\left(\tilde{x}+\Delta \tilde{x}\right)\right)}^{2}}.$$

These equations can be solved for $\tilde{x}$ and $V_{pi}$ using measured $\Delta \tilde{x}$ values.

Extracting $\omega_{0}$ and $Q_{0}$ from measurements is complicated by the non-linear behavior of the electrostatic force. If the relay is moving only a short distance while affected by an electrostatic force, the electrostatic force can be represented as a constant force and a nonlinear, electrostatic contribution to the spring constant, $k_{es}$. The electrostatic contribution to the spring constant can be found by taking a derivative of the electrostatic force with respect to the gate displacement:(5)$$k_{es}=\frac{dF_{el}}{dx}=-\epsilon_{0}V^{2}\frac{A}{{\left(g-x\right)}^{3}}=-k\frac{8}{27}\frac{V_{gb}^{2}}{V_{pi}^{2}}\frac{1}{{\left(1-\tilde{x}\right)}^{3}},$$

Combining the mechanical spring constant with the electrostatic spring contribution given in Equation (Equation 5) gives an effective spring constant for a small displacement around the actuator’s equilibrium point, given by
(6)$$k_{eff}=k\left(1-\frac{8}{27}\frac{V_{gb}^{2}}{V_{pi}^{2}}\frac{1}{{\left(1-\tilde{x}\right)}^{3}}\right).$$

This effective spring constant produces an effective quality factor, $Q_{eff}$ and an effective resonant frequency, $\omega_{eff}$. Expressions for $Q_{eff}$ and $\omega_{eff}$ can be found by substituting (Equation 6) into the expressions for $Q_{0}$ and $\omega_{0}$:(7)$$Q_{eff}^{2}=\frac{mk_{eff}}{b^{2}}=Q_{0}^{2}\left(1-\frac{8}{27}\frac{V_{gb}^{2}}{V_{pi}^{2}}\frac{1}{{\left(1-\tilde{x}\right)}^{3}}\right),$$
(8)$$\omega_{eff}^{2}=\frac{k_{eff}}{m}=\omega_{0}^{2}\left(1-\frac{8}{27}\frac{V_{gb}^{2}}{V_{pi}^{2}}\frac{1}{{\left(1-\tilde{x}\right)}^{3}}\right).$$

The expression $\frac{8}{27}\frac{V_{gb}}{V_{pi}}\frac{1}{{\left(1-\tilde{x}\right)}^{3}}$ is called the voltage-displacement term.

The effective natural frequency and quality factor are extracted from measurements by observing the period of oscillation of the step response, $T_{osc}$, and the amplitude of successive velocity peaks, ${\dot{\tilde{x}}}_{max,i}$, where *i* is an index. Figure 2 shows these parts of the velocity curve. Standard second order differential modeling can be used to derive equations that relate $T_{osc}$ and ${\dot{\tilde{x}}}_{max,i}$ to $\omega_{eff}$ and $Q_{eff}$:(9)$$T_{osc}=\frac{2\pi }{\omega_{eff}\sqrt{1-\frac{1}{4Q_{eff}^{2}}}},$$
(10)$${\dot{\tilde{x}}}_{max,i}={\dot{\tilde{x}}}_{max,0}\exp\left(-\frac{\omega_{eff}}{2Q_{eff}}T_{osc}i\right),$$
where ${\dot{\tilde{x}}}_{max,0}$ is the height of the first peak in the velocity measurement.

### 2.5. Contact Model

Careful examination of LDV measurements when the drain and source make contact, which can be seen in Figure 3, indicate that the free-space model does not describe this regime well. The free space model fails to accurately predict $\omega_{eff}$ and $Q_{eff}$ when the drain and the source make contact. The techniques in Section 2.4 can be used to find $\omega_{eff}$ and $Q_{eff}$ for the closing transient in Figure 3, and these extracted quantities are called $\omega_{con}$ and $Q_{con}$. Though Equations (Equation 7) and (Equation 8) indicate that $\omega_{eff}$ and $Q_{eff}$ should decrease with high $V_{gb}$ and $\tilde{x}$, the extraction process indicates that $\omega_{con}$ and $Q_{con}$ are higher than free-space $\omega_{eff}$ and $Q_{eff}$. This behavior can be captured in simulation by updating the model presented in Equation (Equation 2):(11)$$\frac{\ddot{\tilde{x}}}{\omega {\left(\tilde{x}\right)}^{2}}+\frac{\dot{\tilde{x}}}{\omega \left(\tilde{x}\right)Q\left(\tilde{x}\right)}+\tilde{x}=\frac{4}{27}\frac{V_{gb}^{2}}{V_{pi}^{2}}\frac{1}{{\left(1-\tilde{x}\right)}^{2}}+\exp\left(p_{con}\left(\tilde{x}-g_{d}/g\right)\right).$$

Several terms have been added to this updated model: $\exp\left(p_{con}\left(\tilde{x}-g_{d}/g\right)\right)$ is a force, called $F_{con}$ in future equations, that represents contact with the surface, $p_{con}$ is a tunable convergence parameter that is discussed below, and $\omega (\tilde{x})$ are $Q(\tilde{x})$ are a natural frequency and a quality factor that change as a function of $\tilde{x}$.

There are multiple ways to implement $F_{con}$. The simplest implementation is a piece-wise function that is large when $\tilde{x}>g_{d}$, but piece-wise functions can cause convergence issues for simulators. Another implementation is using high order polynomials to accurately represent a three dimensional integral of Lennard-Jones potentials, and that implementation appears in [8]. Such a high order polynomial can include a third order term to represent an attractive Van Der Waals force. Instead of these other implementations, $F_{con}$ is implemented as an exponential here for three reasons: it is smooth, monotonic and continuously-differentiable; Verilog-A has a limexp function that provides a safe representation of exponentials to simulators; and the proportionality constant, $p_{con}$, is a simple parameter to tune if the model is not converging. $p_{con}$ was set to a large value (500) to make $F_{con}$ approximate a step in force.

The contact model changes $\omega_{0}$ and $Q_{0}$ to new values, $\omega (\tilde{x})$ and $Q(\tilde{x})$, when the drain and source electrodes are in contact. These new values reflect changes in the spring constant, *k*, and the damping constant, *b*. These changes can be caused by a variety of effects: squeeze film damping (which may explain the “bump” in the closing transients in Figure 3 based on preliminary analysis from [20]), energy lost to impact with the surface, the increased deflection of the gate relative to the drain and source electrodes, etc. This model does not attempt to capture each of those phenomena in detail. Instead, the model is designed to mimic behavior extracted from the LDV by using a transition function to modify $\omega_{0}$ and $Q_{0}$ as $\tilde{x}$ becomes large:(12)$$\omega \left(\tilde{x}\right)=\omega_{0}+\left(\omega_{con}-\omega_{0}\right)\cdot \frac{1}{2}\left(1+\tanh(p_{\omega }\left(\tilde{x}-f_{\omega }g_{d}/g\right))\right),$$
(13)$$Q\left(\tilde{x}\right)=Q_{0}+\left(Q_{con}-Q_{0}\right)\cdot \frac{1}{2}\left(1+\tanh(p_{Q}\left(\tilde{x}-f_{Q}g_{d}/g\right))\right).$$

These functions use a hyperbolic tangent to approximate a smooth change in the value of $\omega_{0}$ and $Q_{0}$. The hyperbolic tangent is described by new convergence parameters: $p_{\omega }$ and $p_{Q}$ describe how quickly $\omega_{0}$ and $Q_{0}$ change as $\tilde{x}$ changes, and $f_{\omega }$ and $f_{Q}$, which are fractions between 0 and 1, describe how much $\tilde{x}$ must displace before the $\omega_{0}$ and $Q_{0}$ begin to change. $p_{\omega }$ and $p_{Q}$ were set to 100, and $f_{\omega }$ and $f_{Q}$ were set to 0.9.

## 3. Results

### 3.1. Velocity and Displacement Data

Several trials of the experiment in Section 2.3 were conducted using different body voltages, and LDV produced plots of the transient velocity of the MEM relay for each trial. Integrating the velocity plots produced plots of displacement vs. time, which are pictured Figure 4. LDV does not produce a signal that is perfectly centered at 0 mm/s, so it was necessary to correct this artefact during integration. The curves in Figure 4 have a correction factor of c(t−t0) added to each point of the displacement curve, where $t_{0}$ is the time at the start of the measurement, *t* is time, and *c* is a correction factor that was extracted from the slope of the uncorrected position data.

### 3.2. Parameter Extraction

The first step in parameter extraction was calculating $\omega_{eff}$, $Q_{eff}$ and $\Delta \tilde{x}$ for each of the transient responses. $\omega_{eff}$ and $Q_{eff}$ varied slightly from cycle to cycle because the amplitude of the oscillation decayed, which affected the quality of the small-displacement approximation that was used to calculate the electrostatic spring. The mean of the $\omega_{eff}$ and $Q_{eff}$ values from the first four cycles was used for the analysis. $\omega_{con}$ and $Q_{con}$ were calculated from transient responses where the drain and source made contact, but $\omega_{con}$ and $Q_{con}$ are not included in calculations of $\omega_{0}$ and $Q_{0}$.

The second step of parameter extraction was finding $V_{pi}$ using the measured value for $\Delta \tilde{x}$, the known values for $V_{gb,open}$ and $V_{gb,closed}$, and Equations (Equation 3) and (Equation 4). In addition to a $V_{pi}$ estimate, solving these equations produced a $\tilde{x}-V_{gb}$ curve, pictured in Figure 5. The extracted $\tilde{x}-V_{gb}$ points are compared against an analytical estimate of the position that was found by using Equation (Equation 2) using process data. The extracted curves match analytical expectations closely, deviating only at high $V_{gb}$ values. At those values, $\tilde{x}$ is relatively large, so fringing capacitance, which is not modeled, may be increasing the electrostatic force.

The final step of parameter extraction was finding $\omega_{0}$ and $Q_{0}$. The squares of the extracted $\omega_{eff}$ and $Q_{eff}$ were plotted against the voltage-displacement term from Equations (Equation 7) and (Equation 8). Those plots are pictured in Figure 5. Equations (Equation 7) and (Equation 8) show that $\omega_{eff}^{2}$ and $Q_{eff}^{2}$ both have a linear relationship with the voltage-displacement term, which is reflected in the figure. Linear fits to the data points in the plots for $\omega_{eff}^{2}$ and $Q_{eff}^{2}$ are used to calculate $\omega_{0}^{2}$ and $Q_{0}^{2}$, which are the intercepts of the $\omega_{eff}^{2}$ and $Q_{eff}^{2}$ fits.

### 3.3. Model Validation

The extracted parameters were used in a Verilog-A implementation of Equation (Equation 2), and the Verilog-A model was simulated using hSpice. In the simulation, the Verilog-A model was connected as pseudo-inverter using the same parameter values described in Section 2.3. The model’s simulated velocity is compared with LDV results in Figure 6, which overlays simulated and measured velocity data. Though the measured and simulated velocity data in Figure 6 appear to agree well, figures of merit were necessary to compare the simulation to the measurement. Two such figures of merit were selected: frequency of oscillation and the peak velocity. Figure 6 shows the error in frequency of oscillation and peak velocity as $V_{gb}$ is varied. The error in these figures of merit is never larger than 10%.

## 4. Conclusions

A normalized model of a MEM relay was developed and the parameters of that model were extracted from LDV measurements. The model’s simulated velocity matched measurements over a wide range of operating voltages even though the free-space model only had four parameters.

This improved model could immediately benefit simulations and analysis in publications that follow in the rich tradition of using relay simulations to explore circuit designs, as in [3,8,9,10]. The low complexity and good convergence properties of the model make it particularly suitable for simulating large circuits, as in [10].

This model is tightly tied to parallel plate electrostatic actuators because it relies so heavily on the pull-in voltage as a normalizing term. A natural path for future work is extending this normalization procedure to other capacitance-vs-displacement profiles. Such work could also account more accurately for fringing capacitance, an outstanding inaccuracy in this model at high $V_{gb}$.

## Figures and Tables

**Figure 1 micromachines-13-01728-f001:**
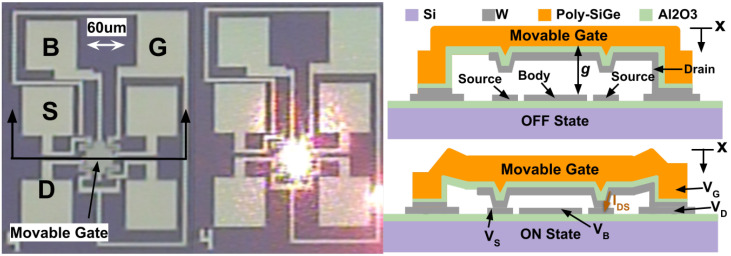
(**left**) A microscope image of the microelectromechanical (MEM) relay device. Body (B), gate (G), source (S), and drain (D) terminals are labeled. (center) A microscope image of the relay showing the laser spot used for laser Doppler vibrometry (LDV). (**right**) A cross-section of the device that indicates the terminals and the signals applied during operation: $V_{g}$, $V_{d}$, $V_{s}$ and $V_{b}$. The cross-section shows that when a voltage difference is applied between the Gate and Body, the device is closed and current ($I_{ds}$) can flow. The actuation gap, *g*, is also labeled.

**Figure 2 micromachines-13-01728-f002:**
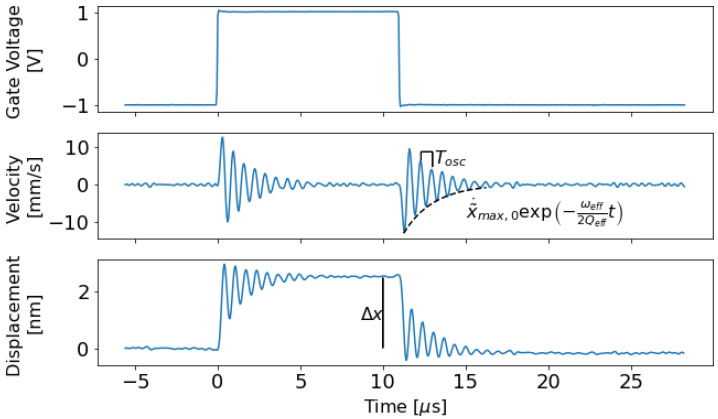
Data collected from a LDV trial. Because the body is at a high, positive bias Voltage, the rising edge of the gate Voltage curve causes the relay to open while the falling edge causes the relay to close. Velocity measured by LDV and displacement calculated by integrating the measured velocity are displayed. The velocity noise deviation is 790 μm/s according to the LDV data sheet: it is given by a noise density of 0.5 μm/s/$\sqrt{\mathrm{Hz}}$ times the square root of the bandwidth, $\sqrt{2.5\mathrm{MHz}}$ [19]. This is higher than the deviation observed in the measured velocity curve above, and may represent a worst case measurement. The LDV displacement resolution, estimated by integrating the velocity noise deviation over the sampling window of 50 ns, is 40 pm.

**Figure 3 micromachines-13-01728-f003:**
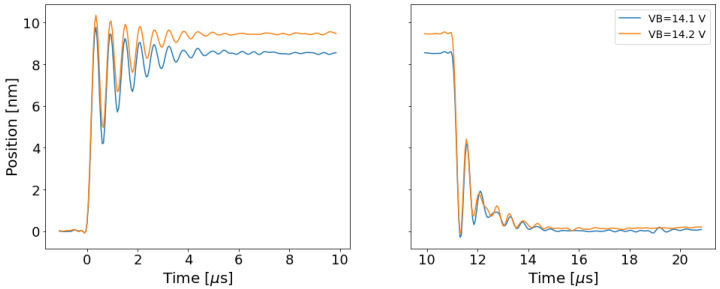
Displacement of the relay during transients that make contact with the surface. $V_{B}$ is higher during these measurements to allow the relay to make contact. The bump in the contacting behavior during the closing transient reflects additional displacement of the gate structure of the relay after the drain and source are in contact, and that displacement may be influenced by squeeze film damping. (**left**) Displacement when the relay is openining. (**right**) Displacement when the relay is closing and making contact.

**Figure 4 micromachines-13-01728-f004:**
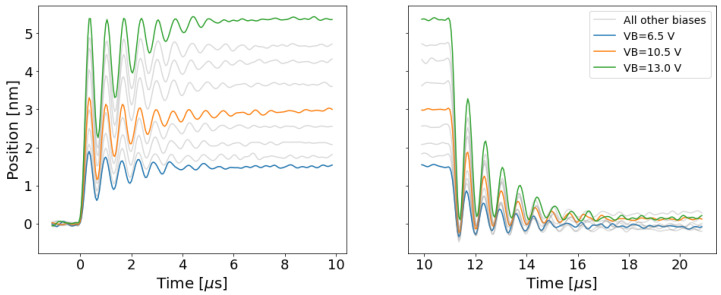
Plots of the displacement of a relay during the transient at each value of $V_{b}$. Positive displacement is measured towards the LDV laser source, so more positive values of displacement are farther from the substrate. (**left**) A plot of the displacement transient when the relay is opening. (**right**) A plot of the displacement transient when the relay is closing.

**Figure 5 micromachines-13-01728-f005:**
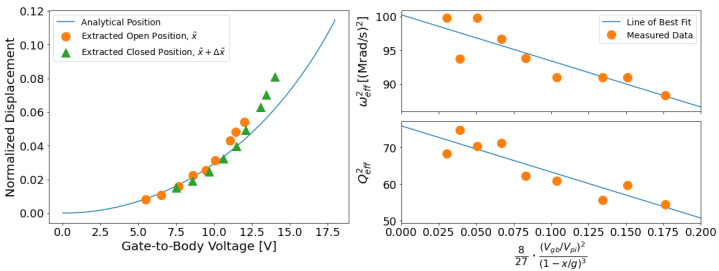
(**left**) A plot of the extracted position before and after closing for a given Gate-to-Body voltage. The incremental displacement, Δx, was found by integrating the velocity curves and used with $V_{gb}$ in the open and closed positions to solve a system of equations for $x_{open}$ and $V_{pi}$. (**right**) Measurements of $Q_{eff}$ and $\omega_{eff}$ used to extract $\omega_{0}$ and $Q_{0}$ from collected data using a linear fit. The true natural frequency and true damping ratio are the y-intercepts of the linear fit.

**Figure 6 micromachines-13-01728-f006:**
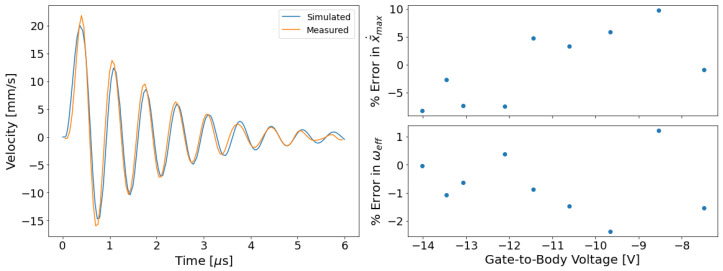
Simulations of the behavioral model developed in this work were compared against velocity measurements. (**left**) A plot of the measured and simulated velocity responses overlaid. The applied body voltage was 12.8 V and the gate voltage was stepped from −1 V to 1 V. (**right**) A summary of the error in figures of merit—frequency of oscillation and peak velocity—in many comparisons between simulated and measured velocity.

## Data Availability

Not applicable.

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
