# Peer review of "A Normalized Model of a Microelectromechanical Relay Calibrated by Laser-Doppler Vibrometry"

_micromachines, 2022, doi:10.3390/mi13101728_

Round 1

Reviewer 1 Report

The authors illustrate an identification method of mechanical structure within a microelectromechanical relay by using laser-doppler vibrometry. The experimental results agree well with the calculation. However, I still have some comments and questions. The responses of my questions would improve the paper scientific merits and engineering values.

1. Section 2.2 is model introduction part, and eq.1 and eq.2 are the identification model. Just as the authors' explanation, the model does not consider the contact and associated bounces behavior. Therefore, the so-called model is only a beam structure with two fixed ends conditions. I admit the method is available and the results is accurate.  However, the value of such model is low. The contact bounce behavior and bounce duration resulted by the rated excited voltage is the significate content. It could not be ignored. 

2. Regarding to Fig.2, the displacement resolution obtained by LDV should be added. also, the contact gap of relay should be given.

3. the obtained intermediate parameters results, such as keff, m, omega eff  Tosc should also be given. 

4. Part 4 discussion should be conclusion

Author Response

Thank you for your thoughtful comments.  The largest change in this revision was adding a section to discuss contact dynamics in response to your first comment.  Detailed responses to each of your comments are below.

  1. Section 2.2 is model introduction part, and eq.1 and eq.2 are the identification model. Just as the authors' explanation, the model does not consider the contact and associated bounces behavior. Therefore, the so-called model is only a beam structure with two fixed ends conditions. I admit the method is available and the results is accurate. However, the value of such model is low. The contact bounce behavior and bounce duration resulted by the rated excited voltage is the significate content. It could not be ignored.

Section 2.5 has been added to the paper to discuss the extension of our methods to modeling contact bouncing behavior.

  1. Regarding to Fig.2, the displacement resolution obtained by LDV should be added. also, the contact gap of relay should be given.

The size of the gap has been added to section 2.1 paragraph 1.  The size of the contact gap has been added to section 2.1 paragraph 3.  Finally, the caption of figure 2 has been augmented with several calculations of the velocity and displacement resolution of the LDV.

  1. the obtained intermediate parameters results, such as keff, m, omega eff Tosc should also be given.

Several of these parameters are unavailable because we are only able to measure omega and Q, which accurately describe the normalized model, but which can’t be reverse engineered to find m.  This model does not require a value for m, and we do not have a way to measure m accurately, so we don’t report it. Because we don’t have an accurate value for m, we cannot accurately calculate keff, so we don’t report it.  Omega_eff is reported in Figure 5.  Because omega_eff has a 1:1 mapping with Tosc (which is given in Equation 9), and because figure 5 is already information-dense, we don’t report Tosc.

  1. Part 4 discussion should be conclusion

Done.

Reviewer 2 Report

I read ‘A normalized model of a microelectromechanical relay calibrated by laser-Doppler vibrometry’ carefully. This paper presents the authors’ model and some results. It includes novel, clear, and coherent contents and, then, this paper should be published in the journal ‘Micromachines,’ as is; however, Editor should check similarity before decisions.

 I have only a few small comments: If I were an author, I would make modifications in Fig. 3 and 4. The titles of Fig. 3 top, ‘Opening (Closing) Transients,’ should be removed. The name of the abscissa in Fig. 4 (left) like 8/27… should be larger. Chapter 4. Discussion does not include significant discussion and this should be ‘Conclusion’ or ‘Summary.’

Author Response

Thank you for your thoughtful comments on the style of the paper.  They have been incorporated into the paper.  An accounting of how we responded to each comment is below.

  1. in Fig. 3 and 4. The titles of Fig. 3 top, ‘Opening (Closing) Transients,’ should be removed.

    Done

  2. The name of the abscissa in Fig. 4 (left) like 8/27… should be larger.

     Done, font size increased for the voltage-displacement equation

  3. Chapter 4. Discussion does not include significant discussion and this should be ‘Conclusion’ or ‘Summary.’

    Done, changed to conclusion.

Round 2

Reviewer 1 Report

I have read the revised manuscript.
It can be accepted as the present style